# Skeleton-Based Spatio-Temporal U-Network for 3D Human Pose Estimation in Video

**DOI:** 10.3390/s22072573

**Published:** 2022-03-28

**Authors:** Weiwei Li, Rong Du, Shudong Chen

**Affiliations:** 1Institute of Microelectronics of the Chinese Academy of Sciences, Beijing 100029, China; liweiwei@ime.ac.cn (W.L.); durong@ime.ac.cn (R.D.); 2University of Chinese Academy of Sciences, Beijing 100029, China

**Keywords:** 3D pose estimation, graph convolutional networks, non-local mechanics, temporal convolutional networks

## Abstract

Despite the great progress in 3D pose estimation from videos, there is still a lack of effective means to extract spatio-temporal features of different granularity from complex dynamic skeleton sequences. To tackle this problem, we propose a novel, skeleton-based spatio-temporal U-Net(STUNet) scheme to deal with spatio-temporal features in multiple scales for 3D human pose estimation in video. The proposed STUNet architecture consists of a cascade structure of semantic graph convolution layers and structural temporal dilated convolution layers, progressively extracting and fusing the spatio-temporal semantic features from fine-grained to coarse-grained. This U-shaped network achieves scale compression and feature squeezing by downscaling and upscaling, while abstracting multi-resolution spatio-temporal dependencies through skip connections. Experiments demonstrate that our model effectively captures comprehensive spatio-temporal features in multiple scales and achieves substantial improvements over mainstream methods on real-world datasets.

## 1. Introduction

Recently, 2D keypoint detection and 3D pose estimation have received more and more attention [1,2,3,4,5,6,7,8]. The difficulty with 3D pose estimation is that multiple 3D poses can be mapped to the same 2D keypoints. Some studies address this ambiguity by modeling temporal information using recurrent neural networks(RNNs) [9,10] or graph convolutional networks(GCNs) [11,12,13]. These studies simply connect temporal dimension features to form a spatio-temporal grid of keypoints for subsequent feature extraction and pose estimation. However, we observe two main problems with these existing methods:the previous methods have difficulty in handling complex long-time sequence action features;the existing approach neglects to consider the compression and fusion of features in both temporal and spatial dimensions for human pose estimation.

First, previous RNN-based and graph-based methods have difficulty in handling complex, long-time sequence action features. A typical approach is to connect keypoints into spatio-temporal sequences based on skeletal structure and use a recurrent neural network [9] or graph convolution network [11] for pose estimation. Some recent studies have proved that temporal convolutional networks have better performance than traditional RNN and other methods in modeling temporal information, such as machine translation [14], language modeling [15], speech generation [16], and speech recognition [17]. Therefore, we employ temporal convolutions to capture long-term pose information for 3D pose estimation tasks.

Second, the existing methods neglect to consider the compression and fusion of features in both temporal and spatial dimensions for human pose estimation. Most of the existing work on human pose recognition focuses on the fusion of spatial skeletal-based keypoint features, such as ESR [12] and ST-GCN [13]. The difficulty lies in the need to aggregate features in both spatio-temporal dimensions to obtain some intermediate representations in local space-time for pose estimation. For example, the concept of "a waving arm” is a semantic fusion representation of multiple keypoints and frames in local space-time. The representation of local spatio-temporal semantics may play a key role in the estimation of the final human pose. The downsampling and upsampling structures work as a bottleneck, encouraging the network to compress feature representations to obtain high-level semantics. The proposed spatio-temporal U-Net architecture helps the network to take full advantage of this bottleneck structure to achieve scale compression and feature squeezing by the downsampling and upsampling of spatio-temporal semantic features.

U-Net structure [18] extracts different resolutions to capture visual patterns or semantics to improve the algorithm performance, and has been successfully applied in many fields such as semantic segmentation, image compression, and denoising. U-Net performs bottom-up processing through the upsampling of feature maps, combined with the underlying high-resolution features, such as the method proposed in the superimposed hourglass network for 2D pose estimation. This U-Net structure only handles different resolutions in the spatial dimension, but cannot handle spatio-temporal information. Inspired by this, as shown in Figure 1, we extend this structure and propose a U-Net structural model scheme for spatio-temporal information. Its purpose is to learn temporal and spatial semantic fusion features for 3D human pose estimation. The data flow is bottom-up, where 2D keypoints are fed into the proposed network and 3D pose estimates are then generated as output.

To the best of our knowledge, we are the first to leverage the temporal convolution and graph convolution to deal with spatio-temporal features of different granularities. Our main contributions are summarized as follows:this work presents a novel spatio-temporal U-Net architecture with a cascade structure of temporal convolution layers and semantic graph convolution layers to gradually integrate the semantic features of local time and space;the proposed structural temporal dilated convolution layer fuses long-time key point sequences in the temporal dimension to eliminate jitter and blur in 3D pose estimation in the single frame case;the proposed semantic graph convolution layer fuses the semantic features of the human body in the spatial dimension with novel graph convolution, pooling, and unpooling layers.

## 2. Related Work

### 2.1. 3D Human Pose Estimation

The 3D Human Pose Estimation task aims to infer 3D body keypoints from a single image. Prior to the success of deep learning, most of the work [19,20,21,22] used feature engineering and modeling on bone and joint mobility to estimate the 3D pose. Next, a convolutional neural network (CNN) method is used for end-to-end 3D pose reconstruction [1,23,24,25]. Unlike previous model-based methods, they estimate 3D pose directly from RGB images without intermediate supervision.

Two-step 3D pose estimation. 3D pose estimation is usually built on top of a 2D pose estimator, first using 2D pose estimation to predict 2D joint positions in image space, and then lifting it to 3D [1,2,3,9,26,27]. Some work [3] shows that predicting 3D pose is relatively straightforward given real 2D keypoints, and the quality of 2D keypoint estimates has a large impact on the final result. Some methods [1,2,28] use both image features and 2D keypoints for 3D pose estimation. Recent work [29] predicts 3D pose by predicting the depth of keypoints. There are methods [30] for 3D pose estimation using prior knowledge about bone length and projection consistency. Some recent studies [31,32] apply transformer networks to human pose estimation tasks. Recently, a differentiable epipolar transformer network in a synchronized and calibrated multi-view setup was proposed [31], enabling the 2D detector to leverage 3D-aware features to improve 2D pose estimation. A spatial-temporal two-stream transformer network [32] is proposed to model dependencies between joints using the Transformer self-attention operator. In addition, the human skeleton can be represented as a directed graph [33] to explicitly reflect the hierarchical relationships among the nodes and leverage varying non-local dependence for different poses by conditioning the graph topology on input poses.

Skeletal-based keypoint feature fusion. GCNs are introduced to learn high-level representations of relationships between nodes based on skeletal graphs. A recent study [11] designed a semantic GCN capable of capturing local and global relationships between human joints for human pose estimation. GCNs are used to learn multi-scale representations [12] to encode human skeletal joints, thereby converting 2D human joints to 3D. Most of these existing studies focus on the analysis of the spatial features of skeleton-based keypoints. Inspired by this, we extend the spatial dimensional fusion method to achieve the fusion of pose features in the spatio-temporal dimension.

Action recognition. Unlike pose estimation, which outputs 3d keypoint coordinates, action recognition directly classifies human behavior. Although the tasks are slightly different, there is also a lot of work in the field of action recognition that focuses on the analysis of the human skeleton structure. A spatial-temporal two-stream transformer network [32] is proposed to model dependencies between joints using the Transformer self-attention operator. Additionally, some work [34] has been done to explore and compare different ways of extracting human pose features, and to extend a TCN-like unit to extract the most relevant spatial and temporal characteristics for a sequence of frames.

### 2.2. Video Pose Estimation

Most of the previous work takes a single frame or a single image as input, and more recent studies use the temporal information of video to disambiguate pose estimation, producing more reliable and robust results. Previous studies [35,36] have used LSTMs to predict 3D poses predicted from a single image. Additionally, an LSTM sequence-to-sequence learning model [37] is introduced to encode 2D pose sequences in videos into fixed-size vectors, which are then decoded into 3D pose sequences. There is also some work on RNN methods that consider prior information on body part connectivity [10]. Some research [13,38,39,40] connects skeleton graphs into keypoint sequences and use GCNs for action recognition. Further, ref. [41] uses a TCN to process pose-encoded sequences, but this method ignores the structural features of the skeleton.

Since none of the existing 3D pose estimation methods consider the representation of features from different granularities of time and space at the same time, we propose a spatio-temporal U-Net model scheme to learn the semantic fusion features of time and space, and perform 3D human pose estimation.

## 3. Skeleton-Based Spatio-Temporal U-Net

As shown in Figure 2, we propose a novel spatio-temporal U-Net scheme to deal with spatio-temporal features of different granularities for 3D human pose estimation in video. The STUNet architecture consists of a cascading structure of structural temporal convolution network(S-TCN) layers and semantic graph convolution network (S-GCN) layers to progressively integrate semantic features in local time and space.

We improve upon the underlying U-Net [18] structure, using skip connections to connect spatio-temporal features from the encoding stage to the decoding stage of each decoder layer. This structure can gradually abstract complex spatio-temporal information to obtain high-level semantic features of pose, and preserve local spatio-temporal information through skip connections. Modeling the 3D pose estimation problem as a U-Net model helps to predict more accurate 3D coordinates through high-level abstractions, and also helps to discover potential relationships between keypoints in temporal and spatial dimensions. The spatio-temporal features of different granularities are extracted and fused in the U-Net structure, which ultimately improves the accuracy of 3D human pose estimation.

### 3.1. Structural Temporal Dilated Convolutional Layer

To improve long-range temporal perception while avoiding excessive increases in training parameters, we employ time-dilated convolutional layers to fuse long-term keypoint sequences in the temporal dimension to alleviate jitter and ambiguity in 3D pose estimation. Dilated convolution is a sparsely structured convolution with a uniform distribution of kernel points and zero padding in between. Suppose there are dilated convolutions of two signals *f* and *h* with lengths *N* and 2M+1, respectively, which can be calculated as:(1)$$\left(f*h_{D}\right)\left[n\right]=\sum_{m=-M}^{M}f\left[D\left(n-m\right)\right]\;h\left[m\right]$$
where *D* represents the dilation factor, and *n* and *m* represent the indices of the signals *f* and *h*, respectively. In Figure 3, we describe the structure of our model in the temporal dimension, whose perceptual domain grows with increasing horizontal layers. In the implementation, we use a similar approach to our previous work [41]. However, the difference is that our S-TCN retains the skeleton information and is able to fuse temporal dimensional features from graph structures of different granularity. The proposed S-TCN yields roughly the same computational cost as conventional convolution while increasing the perceptual field.

### 3.2. Semantic Graph Convolutional Network

The semantic graph convolution layer fuses the semantic features of the human body in the spatial dimension with novel graph convolution, pooling, and unpooling layers. The network consists of a skeletal structure-based graph network layer and a data-dependent non-local layer in series. The structure-based graph layer is used to capture the spatial dimensional human skeletal structure information and progressively pool it into a high-level feature representation. The data-dependent non-local layer is used to analyze the features of long-range nodes since the graph convolution network does not handle long-range relationships well.

#### 3.2.1. Structure-Based Graph Layer

In the classic GCN [13], the graph convolution operation on vertex $v_{i}$ is expressed as:(2)$$f_{out}\left(v_{i}\right)=\sum_{v_{j}\in B_{i}}\frac{1}{Z_{ij}}f_{in}\cdot w\left(l_{i}\left(v_{j}\right)\right)$$
where *v* represents the joint vertex of the skeletal graph and *f* is the feature map. $B_{i}$ represents the convolution sampling region of $v_{i}$, defined as the neighboring vertices $v_{j}$ of the target vertex $v_{i}$. *w* is a weighting function that processes the input values to provide a weight vector. Using a design similar to SemGCN [11], we introduce learnable matrices that transform traditional GCN (Equation 2) as follows:(3)$$f_{out}=\sum_{k}^{k_{v}}W_{k}\left(f_{in}A_{k}\right)⊙M_{k}$$
(4)$$A_{k}=D_{k}^{-\frac{1}{2}}\left({\tilde{A}}_{k}+I\right)D_{k}^{-\frac{1}{2}},D_{ii}=\sum_{k}^{k_{v}}\left({\tilde{A}}_{k}^{ij}+I_{ij}\right)$$
where $k_{v}$ is the kernel size on the spatial dimension, ${\tilde{A}}_{k}$ is the adjacency matrix of the keypoint graph representing connections, *I* is the identity matrix, and $W_{k}$ is the trainable weight matrix.

#### 3.2.2. Graph Pooling and Upsampling

As shown in Figure 4, we divide the body key point nodes into five subsets according to the characteristics of the skeleton structure, and then perform a maximum pooling operation on each subset. Next, the coarsened graph is maximally assembled into a node that contains global information for the entire skeleton. In this process, the skeleton space structure is gradually fused and connected with the corresponding decoded layer in the skip connection of U-Net. Vertex features in the graph of the same granularity are assigned to corresponding vertices in the graph during the upsampling process to fully preserve local spatio-temporal features.

Based on the skeletal structure, the associated neighborhoods are established on the graph to perform the pooling operation, and the semantically similar vertices are clustered together to learn the key representations based on the graph. In this work, we progressively cluster the entire skeleton at each frame according to the structure of human limbs. For the bottom-up process, we use a simple upsampling procedure to copy the features of the vertices in the coarser graph to the corresponding vertices in the fine-grained graph. These higher-level features are concatenated with the lower-level features from skip connections for subsequent processing. Furthermore, temporal connections remain unchanged across different levels of spatio-temporal abstraction.

The fusion of the spatial skeleton contains only two graph pooling processes. In contrast, according to Figure 3, the fusion process of the input time series increases as the length of the series increases. For example, the perceptual domains of 27-frame and 81-frame are fused three and four times, respectively. Therefore, graph pooling layers are selectively inserted into TCN layers, which is related to the granularity of temporal and spatial fusion. We discuss this issue in detail in the subsequent ablation experiments.

#### 3.2.3. Data-Dependent Non-Local Layer

Since the basic GCN has difficulty handling long-distance relationships, we design a data-dependent non-local layer to capture the global and long-distance relationships between joints in the body skeletal map. We follow the non-local [42] concept and define the operation as:(5)$$x_{i}^{l+1}=x_{i}^{l}+\frac{W_{x}}{K}\sum_{j=1}^{k}f\left(x_{i}^{l},x_{j}^{l}\right)\cdot g\left(x_{j}^{l}\right)$$
where $W_{x}$ is the weight, *f* denotes the pairwise function to calculate the affinity between node *i* and other nodes *j*, and *g* is the function of calculating node representation. In implementation, we adopt [42] in a similar way as Equation (Equation 5).

## 4. Experiments and Results

### 4.1. Datasets and Metrics

Datasets. We use the Human3.6M [21] and HumanEva-I [43] datasets for experiments. Human3.6M is the most widely used dataset for 3D pose estimation tasks. It contains 3.6 million images captured in different views by four simultaneous cameras. The dataset consisted of 11 human subjects performing 15 indoor daily activities, such as walking, talking on the phone, sitting, and participating in discussions. The dataset uses a motion capture system to detect precise 3D coordinates and then obtains 2D poses based on the projection of internal and external camera parameters. HumanEva-I is a relatively earlier dataset for 3D human pose estimation. It has a small amount of data with relatively simple pose estimation scenarios. We used the same training and testing split approach as in previous work [10,41,44,45], evaluating multiple subject actions including walking, jogging, and boxing. Following previous work [6,7,8], standard normalization is used to handle the distribution of 2D and 3D poses before data input.

Metrics. There are two standard protocols to evaluate our model on Human3.6M. Following previous work [6,7,8], five subjects (S1, S5, S6, S7, and S8) were used as the training set and two subjects (S9 and S11) were used as the test set, which are evaluated under protocol 1 and protocol 2. As in previous work [3,5,7,12,46], we used two metrics in the dataset Human3.6M to evaluate our method under protocol 1 and protocol 2. Protocol 1 is the metric used is the mean position error per joint (MPJPE), which measures the average Euclidean distance between the ground truth and prediction after alignment of the root joint. Protocol 2 uses the mean per-joint position error (P-MPJPE) after alignment so that it is not affected by rotation and scaling and has better robustness.

### 4.2. Implementation Details

Following previous work [6,7], we use the predicted 2D keypoints released by [41] from the Cascaded Pyramid Network (CPN) as the input of our 3D pose model. Since there is a strong correlation between clips of the same video screen, we sample from different video clips to avoid biased statistics for batch normalization [47].

We trained 100 epochs using the AMSGrad optimizer [48]. An exponential decay learning rate scheme was employed, starting with η=0.001 and applying a shrink factor of α=0.95 per epoch. We also set the batch size and dropout rate to 1024 and 0.2, respectively. The pose data are enhanced by horizontal flipping.

### 4.3. Experimental Results

Learned weighting matrices. The proposed U-Net network contains S-GCN layers in each level. Figure 5 visualizes learned weighting matrices in the network, including the weights in the original map with 17 keypoints and the fusion map with five keypoints. The weight in the upper left is larger than the lower right, which means that the central node has a higher impact than the end node. In other words, the keypoint information is passed through the S-GCN based on the skeleton structure, which proves that the skeleton structure information is fully utilized. In addition, we observe that the weights learn the structural features of the human skeleton. For example, the head, nose, and neck have relatively fixed structural relationships, and the connection weights obtained through training are relatively high. Figure 5 demonstrates that S-GCN correctly resolves the structure of key points of the human skeleton, thus improving the performance of 3D body pose estimation.

Comparison with the state of the art. Table 1 and Table 2 show the results of comparing our proposed STUNet model with other baselines on Human3.6M. We show the performance of our 27-frame, 81-frame, and 243-frame models to compare the performance differences of our models under different perceptual domains. Among them, the bold result is the best. The experiments show that our 243-frame model achieved the best mean error values under protocols 1 and 2. Specifically, we had six tasks reach the optimum under protocol 1 and four tasks outperform the other models under protocol 2. The results show that the model error increases as the number of input frames decreases. Our 81-frame and 27-frame models have about 0.5 mm and 1.9 mm higher average error than the 243-frame model, respectively. The 81-frame and 27-frame models do not outperform the current best results, but are still a relatively competitive result. For protocol 1 in Table 1, our model has an average lead of 0.6 mm compared to the previous best result [49]. Compared to the baseline model [41], our 243-frame model is 6.8 mm and 5.7 mm ahead for the “sitting” and “seated” actions, respectively, indicating that our model is better able to cope with complex situations such as occlusion and overlay. For protocol 2 in Table 2, our method achieves a minimum error of 35.4 mm, which is a reduction of 0.2 mm [49] compared to the best result. It is worth mentioning that our model has a great advantage on highly dynamic action sequences, especially the ”Walk” and ”Walk Together” sequences, which are reduced by more than 4 mm in protocol 1 compared to the base model.

The test results of the HumanEva-I dataset are shown in Table 3, where “-” indicates that no corresponding results are reported in this work. The experiments show that we achieved the best results in seven out of nine sequence tasks for Walk, Jog, and Box. It is worth mentioning that since the HumanEva-I dataset is a relatively easy task, our result metrics still have a slight lead over existing methods. We attribute it to spatio-temporal semantic fusion that smooths dynamic action prediction to improve the error of 3D pose estimation. In general, our model has high performance under various behavioral tasks and multiple evaluation protocols compared to existing models.

To visualize the output of the model, Figure 6 shows the visual qualitative results of our model under multiple action sequences, being the ”Walk”, ”Wait”, ”Pose”, and ”Purch” sequences. We follow our previous work [41] and use the 2D keypoints estimation results of the Cascaded Pyramid Network (CPN) as input. The figure shows that our STUNet model has stable and accurate results on multiple action sequences.

### 4.4. Computational Complexity

As shown in Table 4, we report the number of model parameters and floating point operations (FLOP) and compare them with previous work. The performance comparison under protocol 1 is also shown in Table 4. For the 243-frame model, with a slight increase in parameters and FLOPs, the average error metric of the model reaches the best. In the 27-frame and 81-frame models, the number of parameters and computations are reduced substantially as the acceptance domain is reduced, while also largely maintaining a relatively competitive performance. This is attributed to the fact that we retain the skeleton map information when performing feature fusion to improve accuracy, while the U-Net structure compresses the spatio-temporal information to control the number of parameters.

### 4.5. Ablation Study and Analysis

Granularity of spatio-temporal features. We performed ablation experiments on our models using the Human3.6M dataset. To explore the impact of spatio-temporal fusion at different granularities, we tested the 81-frame and 243-frame models and compared them using the MPJPE metric. The ablation studies are designed to investigate the effects of the order and granularity of temporal and spatial fusion. $N_{1}$ and $N_{2}$ represent the locations of spatial graph pooling operations, respectively, which determine the timing of spatio-temporal feature fusion in the framework. At higher levels of the spatio-temporal dimension, features of specific keypoints are grouped and fused, and some fine-grained information is lost. This fine-grained information is concatenated in the subsequent upsampling process through skip connections. The larger the values of $N_{1}$ and $N_{2}$, the later the spatio-temporal features are fused, which means that more layers of the model process features in the fine-grained spatio-temporal dimension. As a result, we achieve the best results at (1,3) and (2,4), respectively. The best results for $N_{1}$ and $N_{2}$ for both frames 81 and 243 take intermediate values. It is worth mentioning that Table 5 shows that our model is not sensitive to the hyper-parameter values of $N_{1}$ and $N_{2}$ and the effect on the results is within 1mm, indicating its strong robustness.

U-Net architecture. This work presents a novel STUNet architecture with a cascade structure of temporal convolution layers and graph convolution layers to gradually integrate the semantic features of local time and space. Skip connections combine coarse-grained feature embeddings from the decoder sub-network with fine-grained spatio-temporal feature embeddings from the encoder sub-network to boost the final 3D pose estimation performance. Experiments demonstrate that skip connections are effective in recovering and fusing spatio-temporal features of different granularities.

Spatio-temporal feature fusion. Our approach performs feature fusion and compression in both the temporal and spatial dimensions. The proposed structural temporal dilated convolution layer can retain spatial structured graph information while fusing long-time key point sequences in the temporal dimension to eliminate jitter and blur in 3D pose estimation in the single frame case. The semantic features of the human body structure are processed by semantic graph convolution, which leads to a significant improvement in the accuracy of our model compared to the base model.

Optimization of multi-frame input. Models with multiple frames of input are usually more difficult to practice for specific scenarios than single frames. However, we applied dilated TCN to handle temporal dimensional features, which provides the possibility of implementing application optimization. Similar to other dilated TCN-based approaches [41], since the time-dimensional features are processed hierarchically, features at different moments can be subsequently reused as long as they are computed once at the inference time. Therefore, the input of 243 frames only affects the start and end of model inference, while the runtime inference is able to be optimized for parallelism and reuse, maintaining a good performance.

## 5. Conclusions

In this paper, we address the problem that existing methods lack effective means to extract dynamic complex features from spatio-temporal structures. We propose a novel spatio-temporal U-Net scheme to deal with spatio-temporal features in multiple scales for 3D human pose estimation in video. The proposed STUNet architecture consists of a cascade structure of semantic-structural graph convolution layers and temporal dilated convolution layers, progressively extracting and fusing the spatio-temporal semantic features from fine-grained to coarse-grained. Our method achieves competitive performance on 3D body pose estimation benchmarks. It is experimentally demonstrated that the U-shaped network structure optimizes 3D pose estimation by the downscaling and upscaling of spatio-temporal fusion features. The experiments also show the importance of the representation of spatio-temporal features at different granularities in the pose recognition task, which opens up many possible directions for future work. For example, how to get rid of manual hard rules and apply a graph pooling method with trainable rules [50,51] to automatically fuse spatio-temporal features of different granularities for pose estimation tasks.

## Figures and Tables

**Figure 1 sensors-22-02573-f001:**
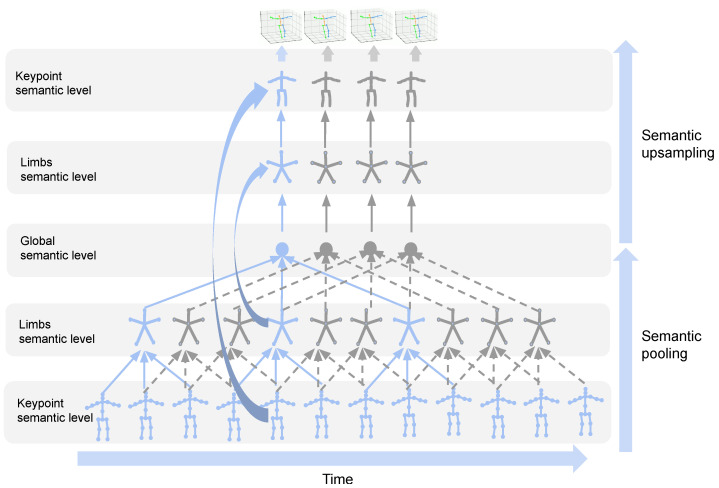
Illustration of the proposed skeleton-based spatio-temporal U-Net. In the semantic pooling stage, spatio-temporal semantic features are gradually compressed and fused into different granularities. In the semantic upsampling phase, spatio-temporal features are decoded and multi-resolution spatio-temporal dependencies are abstracted by skipping connections in the U-Net structure.

**Figure 2 sensors-22-02573-f002:**
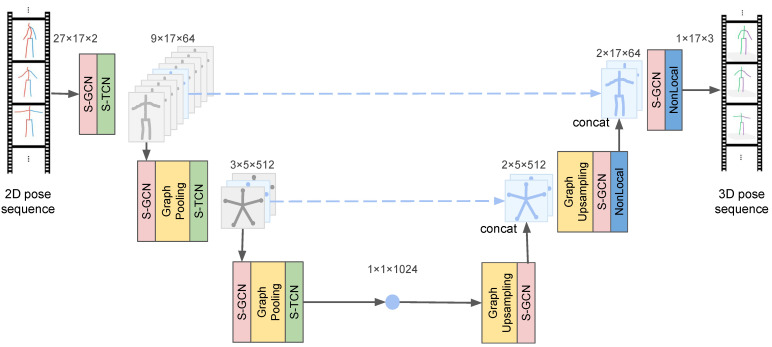
Overview of the proposed spatio-temporal U-Net scheme. The STUNet architecture consists of a cascade structure of semantic-structural graph convolution network(S-GCN) layers and structural temporal convolution network (S-TCN) layers to progressively integrate semantic features in local time and space. Taking 27 frames of input as an example, the model contains two layers of graph pooling in the spatial dimension and three layers of TCN compression in the temporal dimension.

**Figure 3 sensors-22-02573-f003:**
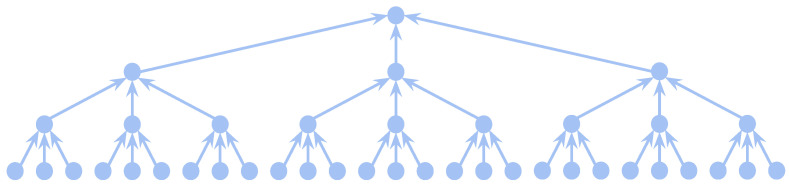
Data flow in the proposed S-TCN model, from bottom input to top output.

**Figure 4 sensors-22-02573-f004:**
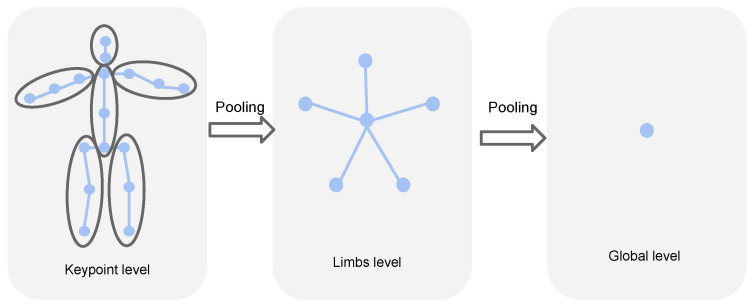
The defined hierarchical graph pooling strategy for the human body.

**Figure 5 sensors-22-02573-f005:**
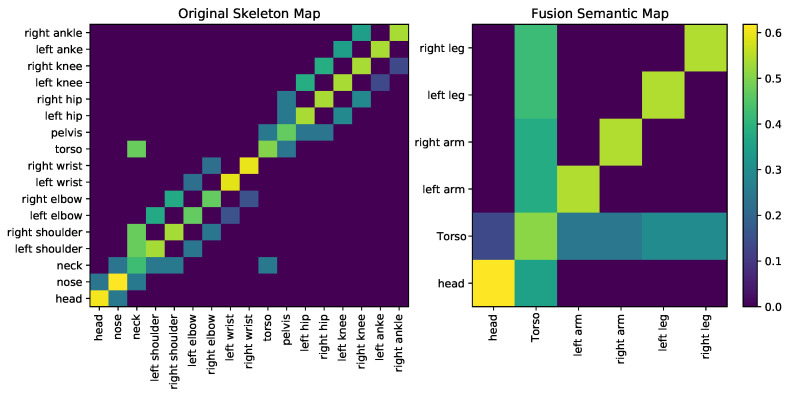
Visualization of learned weighting matrices, *M*, of S-GCN in the network.

**Figure 6 sensors-22-02573-f006:**
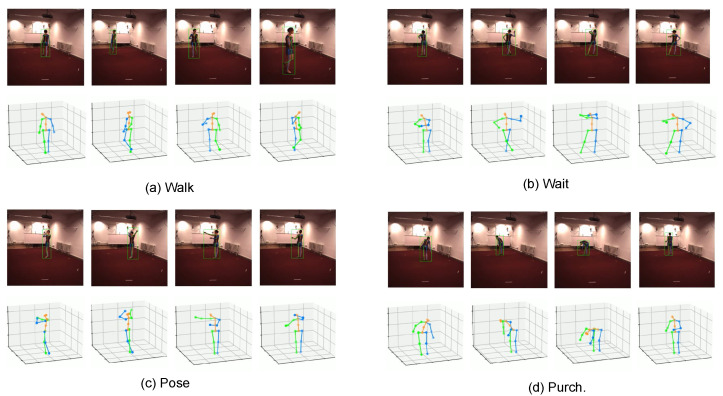
The visualized qualitative results of 3D pose estimation in video.

**Table 1 sensors-22-02573-t001:** Reconstruction error on Human3.6M under protocol 1. Results are in millimeters.

	Dir.	Disc.	Eat	Greet	Phone	Photo	Pose	Purch.	Sit	SitD.	Smoke	Wait	WalkD.	Walk	WalkT.	Avg
Pavlakos et al. [1]	67.4	71.9	66.7	69.1	72.0	77.0	65.0	68.3	83.7	96.5	71.7	65.8	74.9	59.1	63.2	71.9
Fang et al. [5]	50.1	54.3	57.0	57.1	66.6	73.3	53.4	55.7	72.8	88.6	60.3	57.7	62.7	47.5	50.6	60.4
Pavlakos et al. [6]	48.5	54.4	54.4	52.0	59.4	65.3	49.9	52.9	65.8	71.1	56.6	52.9	60.9	44.7	47.8	56.2
Yang et al. [7]	51.5	58.9	50.4	57.0	62.1	65.4	49.8	52.7	69.2	85.2	57.4	58.4	43.6	60.1	47.7	58.6
Luvizon et al. [8]	49.2	51.6	47.6	50.5	51.8	60.3	48.5	51.7	61.5	70.9	53.7	48.9	57.9	44.4	48.9	53.2
Hossain et al. [9]	48.4	50.7	57.2	55.2	63.1	72.6	53.0	51.7	66.1	80.9	59.0	57.3	62.4	46.6	49.6	58.3
Lee et al. [10]	40.2	49.2	47.8	52.6	50.1	75.0	50.2	43.0	55.8	73.9	54.1	55.6	58.2	43.3	43.3	52.8
Pavllo et al. [41]	45.9	48.5	44.3	47.8	51.9	57.8	46.2	45.6	59.9	68.5	50.6	46.4	51.0	34.5	35.4	49.0
Cai et al. [12]	44.6	47.4	45.6	48.8	50.8	59.0	47.2	43.9	57.9	61.9	49.7	46.6	51.3	37.1	39.4	48.8
Yeh et al. [44]	44.8	46.1	43.3	46.4	49.0	55.2	44.6	44.0	58.3	62.7	47.1	43.9	48.6	32.7	33.3	46.7
Xu et al. [45]	**37.4**	43.5	42.7	**42.7**	46.6	59.7	**41.3**	45.1	**52.7**	**60.2**	45.8	**43.1**	47.7	33.7	37.1	45.6
Liu et al. [49]	41.8	44.8	**41.1**	44.9	47.4	**54.1**	43.4	42.2	56.2	63.6	45.3	43.5	45.3	31.3	32.2	45.1
Ours (27 frames)	43.5	44.8	43.9	44.1	47.7	56.5	44.0	44.2	55.8	67.9	47.3	46.5	45.7	33.4	33.6	46.6
Ours (81 frames)	42.6	43.6	42.8	43.1	**46.1**	54.6	43.3	42.4	53.5	63.2	45.8	44.2	44.9	31.9	32.0	45.0
Ours (243 frames)	41.9	**43.1**	42.3	42.9	46.3	54.2	42.9	**41.8**	53.1	62.8	**45.3**	43.9	**43.4**	**31.2**	**31.8**	**44.5**

**Table 2 sensors-22-02573-t002:** Reconstruction error on Human3.6M under protocol 2. Results are in millimeters.

	Dir.	Disc.	Eat	Greet	Phone	Photo	Pose	Purch.	Sit	SitD.	Smoke	Wait	WalkD.	Walk	WalkT.	Avg
Martinez et al. [3]	39.5	43.2	46.4	47.0	51.0	56.0	41.4	40.6	56.5	69.4	49.2	45.0	49.5	38.0	43.1	47.7
Sun et al. [4]	42.1	44.3	45.0	45.4	51.5	53.0	43.2	41.3	59.3	73.3	51.0	44.0	48.0	38.3	44.8	48.3
Fang et al. [5]	38.2	41.7	43.7	44.9	48.5	55.3	40.2	38.2	54.5	64.4	47.2	44.3	47.3	36.7	41.7	45.7
Pavlakos et al. [6]	34.7	39.8	41.8	38.6	42.5	47.5	38.0	36.6	50.7	56.8	42.6	39.6	43.9	32.1	36.5	41.8
Yang et al. [7]	**26.9**	**30.9**	36.3	39.9	43.9	47.4	**28.8**	**29.4**	**36.9**	58.4	41.5	**30.5**	**29.5**	42.5	32.2	37.7
Hossain et al. [9]	35.7	39.3	44.6	43.0	47.2	54.0	38.3	37.5	51.6	61.3	46.5	41.4	47.3	34.2	39.4	44.1
Pavllo et al. [41]	34.2	36.8	33.9	37.5	37.1	43.2	34.4	33.5	45.3	52.7	37.7	34.1	38.0	25.8	27.7	36.8
Cai et al. [12]	35.7	37.8	36.9	40.7	39.6	45.2	37.4	34.5	46.9	50.1	40.5	36.1	41.0	29.6	33.2	39.0
Xu et al. [45]	31.0	34.8	34.7	**34.4**	36.2	43.9	31.6	33.5	42.3	**49.0**	37.1	33.0	39.1	26.9	31.9	36.2
Liu et al. [49]	32.3	35.2	33.3	35.8	**35.9**	41.5	33.2	32.7	44.6	50.9	37.0	32.4	37.0	25.2	27.2	35.6
Ours (27 frames)	34.3	35.7	34.9	36.6	37.5	42.7	33.1	36.0	44.4	53.7	38.5	33.5	38.4	26.0	28.4	36.9
Ours (81 frames)	33.5	35.1	33.9	36.0	36.9	**42.1**	32.3	34.5	42.9	50.1	37.7	33.0	37.8	25.6	27.6	36.0
Ours (243 frames)	33.3	34.8	**33.6**	35.2	36.3	42.2	32.1	33.7	42.6	49.4	**36.9**	32.8	37.4	**25.1**	**27.2**	**35.4**

**Table 3 sensors-22-02573-t003:** Reconstruction error on HumanEva-I dataset under protocol 2. Results are in millimeters.

		Walk			Jog			Box	
	S1	S2	S3	S1	S2	S3	S1	S2	S3
Pavlakos et al. [6]	22.3	19.5	29.7	28.9	21.9	23.8	-	-	-
Lee et al. [10]	18.6	19.9	30.5	25.7	16.8	17.7	42.8	48.1	53.4
Pavllo et al. [41]	13.9	10.2	46.6	20.9	13.1	13.8	23.8	33.7	32.0
Yeh et al. [44]	15.2	10.3	47.0	21.8	13.1	13.7	22.8	31.8	31.0
Xu et al. [45]	13.2	10.2	29.9	**12.6**	12.3	13.0	**13.2**	18.1	20.4
Liu et al. [49]	13.1	9.8	26.8	16.9	12.8	13.3	-	-	-
Ours (243 frames)	**12.8**	**9.7**	**26.5**	16.0	**12.2**	**12.7**	14.6	**16.9**	**19.3**

**Table 4 sensors-22-02573-t004:** Computational complexity of various models under protocol 1.

Model	Parameters	FLOPs	MPJPE (mm)
Hossain et al. [9]	16.96M	33.88M	58.3
Pavllo (81 frames) et al. [41]	12.75M	25.48M	47.7
Pavllo (243 frames) et al. [41]	16.95M	33.87M	46.8
Ours (27 frames)	14.80 M	29.03 M	45.8
Ours (81 frames)	19.67 M	38.45 M	45.0
Ours (243 frames)	29.58 M	64.84 M	44.5

**Table 5 sensors-22-02573-t005:** Ablation study for our 81-frame and 243-frame model under protocol 1 on Human3.6M.

Frames	$N_{1}$	$N_{2}$	MPJPE (mm)
81	1	2	45.8
81	1	3	45.0
81	2	3	45.3
243	1	3	45.5
243	1	4	45.3
243	2	3	44.9
243	2	4	44.5
243	3	4	44.7

## Data Availability

Not applicable.

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
