# Peer review of "Skeleton-Based Spatio-Temporal U-Network for 3D Human Pose Estimation in Video"

_sensors, 2022, doi:10.3390/s22072573_

Round 1
Reviewer 1 Report
The results and the conclusions are well done presented, the article has also a well-argued stage and supported by comparative tables.
In the same time, it presents the contribution of authors in a domain in a continuous evolution and different approaches, as a result the bibliography is a topical one, but I propose for consultation other more recent articles such as:
- “Skeleton-based Action Recognition via Spatial and Temporal Transformer Networks” - Chiara Plizzaria, Marco Cannicia , Matteo Matteuccia Computer Vision and Image Understanding, elsevier.com Skeleton, https://arxiv.org/pdf/2008.07404.pdf
- “Spatial temporal transformer network for skeleton-based action recognition”, Chiara Plizzari, Marco Cannici, Matteo Matteucci, 2021/1/10, Conferenza International Conference on Pattern Recognition, pp.694-701, Springer;
- “Comparison between Recurrent Networks and Temporal Convolutional Networks Approaches for Skeleton-Based Action Recognition”, Mihai Nan, Mihai Trascau, Adina Magda Florea and Cezar Catalin Iacob Sensors 2021, 21, 2051, https://doi.org/10.3390/ s2106205
https://pdfs.semanticscholar.org/164c/bfe41d475ceda755ecd27245e1496972c804.pdf?_ga=2.190611435.1130606600.1645991394-1071021338.1639419236
It is a proposal to corroborate different aspects regarding the approach on similar topic.
Regarding the instruction of writing the article I notice that Figure 1 is not mentioned in the text and, even if all abbreviations are known, it is advisable to mention them at the end of the paper.
Reviewer 2 Report
The authors proposed a Skeleton-based Spatio-Temporal U-Network for 3D Human Pose Estimation that is very worth studying. However, I don't see high novelty, and there are still some writing problems in the manuscript, which need the author to further improve.
- There are too many language errors, such as “abstracting abstracts” in line 9, "figure" in line 137 and line 156, 'table 3' in line 245 ...
- The logic of the manuscript is poor, and the author does not highlight the main content to be studied, which is very laborious to read.
- The paper lacks novelty. As far as I know, the spatial temporal graph revolutionary networks proposed by the author is an existing method, and u-net is also a widely used network model. The author only splices or reassembles the mature methods or models.
- The experimental results are insufficient. The author only used one dataset for method verification, and the method has not been widely verified.
- “The spatial skeleton is 261 fused in a finer-grained time domain, but the spatial skeleton information will be gradually 262 lost in the higher-level spatiotemporal features." This expression is difficult for me to understand. Do you want to express that with the deepening of network operation, some visual information will be lost?
- The conclusion section is more colloquial. For example, "we believe..."
- The paper lacks discussion on further research in the future, and future research plans should be added in conclusion section.
Reviewer 3 Report
This was a surprisingly wonderful manuscript to review. The writing quality, thoroughness and methodological transparency are commendable. Additionally, while the contribution of this work is relatively incremental, challenging benchmark results is not trivial and the authors have presented improvements in some domains against many state-of-the-art approaches.
My only criticisms are:
- Tables and figures in the results are missing significant information. For example, Tables 1 and 2 do not have units in the caption or table. Figure 5 does not have axis labels or a legend for the colour scheme.
- The most significant results are achieved through overall average performance using 243 frames, but this only outperforms other models in four of 17 tasks. No results are presented for the 17 frame input.
- Computational complexity is discussed as being comparable or better, but this is not at all true for the 243 frame input, which is the only one that outperforms other approaches.
- Discussion could be provided regarding the practicality of 243 frame inputs for many activities
Overall, while the contributions of this work are incremental, I support the quality of the methodology and transparency in results presented by the authors.
Reviewer 4 Report
This work proposes a novel skeleton-based spatio-temporal U-Net(STUNet) scheme, which intends to deal with spatio-temporal features in multiple scales for 3D human pose estimation in video. The study is supported by a sufficient number of sources and its theoretical basis. However, there are still some problems here.
The comments to the authors are as follows:
1.No mention of Figure 1 in the text. Each figure in the paper should be mentioned in the text and explain the relationship or trend of things expressed by the table.
2. Question regardinabbreviations.
First, there are some incomplete words. For example, S-GCN should be written semantic-structural graph convolution network, but here it is written as "semantic-structural graph convolution" (line 116). In fact, all the abbreviations of this word in the article have this error. (S-GCN: semantic-structural graph convolution network; S-TCN: temporal dilated convolution network)
Second, abbreviations should be used when a lengthy word or phrase is mentioned multiple times in the body of the paper. The author uses abbreviations and full writing in all the text. For example, GCN(line 151,178) and Graph convolutional network(line 85,87,90,144). The same phenomenon also like "semantic-structural graph convolution", "spatio-temporal U-Net".
3. In line 45, the author mentioned "bottleneck strategy" but no explanation. The authors should provide more description about the strategy that advances the experimental process.
4. In line 168, "top-down process" should be "bottom-up process".
5. In line 179, "the pairwise function Calculate the affinity between……”, this sentence is unclear.
6. Letter case error. In line 206, "Amsgrad" should be "AMSGrad".
7. In line 256, "243frames" should be "243 frames".
Reviewer 5 Report
- The proposed spatio-temporal U-Net proposes to achieve scale compression and feature squeezing by downscaling and upscaling of Spatio-temporal semantic features. However, the description of Graph Upsampling is not clear. Since downsampling uses max pooling, upsampling should be impossible because max pooling is irreversible.
- In addition, so far this year, many better practices have been proposed, such as the method using Directed Graph Convolution [1], or the method using Transformer [2]. It is recommended to compare or discuss with these methods
Reference
[1] W. Hu, C. Zhang, F. Zhan, L. Zhang, and T.-T. Wong, “Conditional Directed Graph Convolution for 3D Human Pose Estimation” in arXiv preprint, arXiv: 2107.07797, 2021.
[2] Y. He, R. Yan, K. Fragkiadaki and S. -I. Yu, “Epipolar Transformers,” in IEEE/CVF Conference on Computer Vision and Pattern Recognition (CVPR), 2020, pp. 7776-7785, doi: 10.1109/CVPR42600.2020.00780.
Round 2
Reviewer 2 Report
The author gave detailed responses to all problems by I proposed. I suggest to receive it in its present form.
Reviewer 5 Report
The authors have addressed my questions. I have no further questions. I recommend publishing this paper.